# Detergent and Water Recovery from Laundry Wastewater Using Tilted Panel Membrane Filtration System

**DOI:** 10.3390/membranes10100260

**Published:** 2020-09-27

**Authors:** Nafiu Umar Barambu, Derrick Peter, Mohd Hizami Mohd Yusoff, Muhammad Roil Bilad, Norazanita Shamsuddin, Lisendra Marbelia, Nik Abdul Hadi Nordin, Juhana Jaafar

**Affiliations:** 1Department of Chemical Engineering, Universiti Teknologi PETRONAS, Bandar Seri Iskandar 32610, Perak, Malaysia; barambunafiu@gmail.com (N.U.B.); derrickspeter@gmail.com (D.P.); hizami.yusoff@utp.edu.my (M.H.M.Y.); nahadi.sapiaa@utp.edu.my (N.A.H.N.); 2HICoE-Centre for Biofuel and Biochemical Research, Institute of Self-Sustainable Building, Universiti Teknologi PETRONAS, Seri Iskandar 32610, Perak, Malaysia; 3Faculty of Integrated Technologies, Universiti Brunei Darussalam, Jalan Tungku Link, Gadong BE1410, Brunei; norazanita.shamsudin@ubd.edu.bn; 4Department of Chemical Engineering, Faculty of Engineering, Universitas Gadjah Mada, J1. Grafika 2, Yogyakarta 55281, Indonesia; lisendra.m@ugm.ac.id; 5Advanced Membrane Technology Research Centre (AMTEC), Universiti Teknologi Malaysia, Skudai 81310, Johor Bahru, Malaysia; juhana@petroleum.utm.my

**Keywords:** laundry wastewater, membrane fouling, membrane module, water and detergent recovery

## Abstract

Increasing global concern on clean water scarcity and environmental sustainability drive invention in water reclamation technology. Laundry wastewater reclamation via membrane technology faces the challenge of membrane fouling. This paper assesses a tilting-the-filtration-panel filtration system for the treatment of real laundry wastewater filtration aimed for water and detergent reuse. Results showed that the panel tilting significantly improved fouling control and enhanced permeability due to enhanced contact of air bubbles with the membrane surface, which induced continuous detachment of foulant from the membrane surface. The combination of aeration rate and tilting angle resulted in up to 83% permeability enhancement from 109 to 221.4 ± 10.8 (L/m^2^·h·bar). The system also offers 32% detergent recovery. Overall findings suggest that the system offers an attractive approach for both fouling management and detergent recovery and can potentially be applied under a simple setup in which filtration can be driven by gravity/hydrostatic pressure.

## 1. Introduction

Discharge of untreated wastewater is one of the major challenges facing the world today [1,2]. Thus, the United Nations has come up with more strict legislation to strengthen law enforcement agencies regarding environmental sustainability [1]. Industrial laundries are among the largest producers of wastewater [3,4]. Detergents used in laundry operations contain surfactants, which remove dirt/soil from the contaminated textiles. The excess surfactants discharged to the environment rapidly biodegrade by consuming large amounts of bio-available dissolved oxygen, leading to increased chemical oxygen demand (COD) and thus disrupting the ecological system [5].

Conventionally, laundry wastewater is discharged directly to the water bodies or on the land and therefore affects the ecological system [6]. This draws attention to implementation of laundry wastewater discharge and reuse standards (provided in detail elsewhere) [7]. Prior to discharge or reuse of wastewater, environmental standards for discharge and reuse of wastewater have to be satisfied [8]. Therefore, laundry wastewater has to be treated to improve its quality [4]. Membrane technology has been adopted for water and wastewater treatment due its high separation efficiency as well as its economic and environmental advantages [9]. However, one of its downsides is membrane fouling, which disallows maintaining high permeate flux over a prolonged operation [10]. Thereby, membrane fouling complicates the system by increasing both capital and operational expenditure as well as shortening the membrane life-span [11,12]. The necessity to perform regular maintenance cleaning for membrane fouling management renders the system as very demanding in operation [13].

Membrane fouling occurs as a result of particles (foulant) accumulation on the surface and/or within the pores of the membrane [14,15]. The deposited foulant eventually forms a fouling layer (cake layer) imposing an increase in filtration resistance that leads to flux decline. Therefore, to restore the hydraulic performance, the membrane has to be periodically cleaned, often using chemicals such as alkaline and/or acid-based agents. The exposure to the cleaning chemicals not only affects the material life-span, but the secondary pollutant affects the ecosystem as well [16]. In laundry wastewater treatment, soil, greases, and other forms of dirt removed by the detergent act as the foulant materials, affecting the stability of the filtration system [17]. Surfactants are the major detergent ingredient that reduce the surface tension (interfacial tension) by breaking down the interface between water and dirt (oils, greases, soil, clay, etc.) [5,17]. They also hold the dirt in suspension and allow their removal. Water molecules congregate to the hydrophilic end of the surfactants, while dirt and water insoluble materials congregate to its hydrophobic end, which then can accumulate on and/or within the membrane surface during filtration [3,18].

Water and free/excess detergent in laundry wastewater can be seen as potential resources to be reused or recovered, which can reduce the overall amount of detergent and water consumption [8]. Therefore, treatment of laundry wastewater using membrane filtration offers the economic benefit of recovering the excess/free detergent and water apart from the environmental benefit [3]. Excess/free detergent exists because it is difficult to determine the actual amount of detergent that can remove dirt from the clothes [4]; as such, it is often provided in excess [3].

There are many strategies that have proved to be effective for fouling management, such as membrane surface modification [19], pre-treatment [20], membrane patterning [21,22], and hydrodynamic (inducing turbulence flow) [23]. Inducing turbulence flow of the feed near the membrane surface is another option for fouling management. It disrupts foulant agglomeration and accumulation on the membrane surface and is considered as one of the most effective methods for membrane fouling control [24,25,26]. This study explores the use of air bubbles in a submerged filtration set-up as the means of inducing feed turbulence flow near the membrane surface for fouling mitigation coupled with a tilted panel filtration system.

Numerous studies have been reported on the implementation of membrane technology for laundry wastewater treatment, mostly in combination with other process. Most of the reported literature utilizes a two-stage treatment system that involves pre-treatment of the feed to reduce foulant concentration, mostly via a conventional process, followed by membrane filtration. Ciabatti et al. [27] integrated the membrane process with physio-chemical pretreatment stages of sand filter, ozonation, and activated carbon for treatment of laundry wastewater. They reported up to 87%, 98%, and 99% rejection efficiencies for COD, total suspended solid, and turbidity, respectively. The high removal rates could be attributed to a series of pre-treatments prior the membrane process. Shang et al. [20] employed coagulation as pretreatment to mitigate fouling and improved the membrane hydraulic performance. They reported an increase in critical flux from 45 (L/m^2^·h) to 450 (L/m^2^·h) due to the pretreatment stage that reduced the foulant concentration exposed prior to the membrane filtration stage. Recently, Bilad et al. [28] employed a low pressure membrane filtration system in a standalone process of simultaneous detergent and water recovery from laundry wastewater. The study focused on implementation of low-pressure system as well as the method for implementation in a small-scale unit.

Most of literature focuses on water reuse and, in some cases, detergent recovery without considering environmental and economic aspects of the pre-treatment stage [8,20,27,29]. This study focuses on water reuse and detergent recovery by optimizing hydrodynamic conditions for fouling management. In a submerged filtration system, air bubbling allows membrane fouling control by promoting turbulence near the membrane surface. Therefore, it has been continuously investigated to fully harness its potentials in fouling management [30,31,32]. Nonetheless, maximum impact of air bubbling in fouling management is often achieved at a higher aeration rate, which incurs high energy input. However, a newly developed tilted membrane panel system proves to enhance membrane surface–air bubbles interaction at an economic pumping rate [23], which substantially reduced energy input by operating under optimum conditions [33]. 

This paper assesses the effectiveness of a tilted panel in a submerged membrane system for membrane fouling control in the filtration of laundry wastewater for water and detergent recovery purposes. COD, total phosphorous (TP), turbidity, and total nitrogen (TN) of the laundry wastewater and the treated water were analyzed, and the performance of the system was evaluated in terms of permeability and detergent recovery.

## 2. Materials and Methods

### 2.1. Membrane Preparation and Characterization

The filtration tests were performed using a microfiltration membrane produced via the phase inversion process using a system comprising polyvinylidene fluoride (PVDF, 300 kDa, Arkema, PA, USA) as the polymer, dimethylacetamide (DMAC, Sigma-Aldrich, St. Louis, MO, USA) as the solvent, and demineralized water as the non-solvent. Polyethylene glycol (PEG, 10,000 kDa, Sigma-Aldrich, St. Louis, MO, USA) and lithium chloride (LiCl, Sigma-Aldrich, St. Louis, MO, USA) were used as additives in the dope solution. The casting solution consisted of 15 wt % of PVDF, 1 wt % of PEG, and 0.1 wt % LiCl in 83.9 wt % DMAC. The homogeneous solution was degassed to release any entrapped bubble and was casted using 0.22 mm casting thickness on a nonwoven support (Novatexx 2471, Freudenberg-Filter, Weinheim, Germany) to minimized membrane shrinkage. The cast film was immediately immersed in a nonsolvent bath for the phase inversion process. The resulting membrane was kept in a fresh non-solvent until usage. After characterization, structure, contact angle (^o^), thickness (mm), mean pore size (µm), and clean water permeability (L/m^2^ h bar) of the membrane sample were asymmetric, 80.2 ± 2.2, 166.5 ± 3, 0.33 ± 0.1, and 860 ± 62, respectively. Microfiltration membrane was opted because of its low intrinsic resistance, allowing the filtration system to operate under low transmembrane pressure simply driven by gravity. Such a simple system can be implemented in small-scale laundry industries.

### 2.2. Laundry Wastewater and Detergent Recovery Test

The laundry wastewater sample used as the filtration feed was taken from the first wash cycle of a commercial laundry shop in Seri Iskandar, Perak, Malaysia. The sample was kept in a dark and cold room (4 °C) when not in use to restrict possible microorganism growth, which could contribute to biofouling. Table 1 presents the feed sample characteristics used for the filtration tests. Hach-Lange test kits (Colorado, USA) were used to analyze the feed sample and the permeate COD, TN, and TP, while a turbidity meter was used for turbidity analysis. The whiteness index measurements of the cloth washed with fresh feed and reuse feed were determined as per American Society for Testing and Materials (ASTM E313-05) to indirectly quantify the detergent used. ASTM E313-05 method correlates the detergent quantity with bases on the specimen color, e.g., yellowness or whiteness of white and near-white or colorless viewed in daylight. 

For the filtration test, the membrane was placed into a plate and frame module panel with an effective area of 140 cm^2^ (10 cm × 14 cm). The membrane was compacted by filtration of clean water for an hour prior to laundry wastewater filtration. Equations (1)–(3) were used to evaluate permeate flux *(J),* permeability *(L),* and rejection (*R*) of the membrane, respectively.
(1)$$J=\frac{V}{At}$$
(2)$$L=\frac{J}{∆P}\;\;$$
(3)$$R=\;\frac{C_{f}-C_{p}\;}{C_{f}}\;\times 100$$
where *V* is volume of the permeate (*L*) collected at time *t* (h) over effective area *A* (m^2^) and at transmembrane pressure Δ*P* (bar). *C_f_* and are *C_p_* the concentration of the detergent in the feed and the permeate (mg/L), respectively.

### 2.3. Experimental Set-Up

Membrane performance was evaluated at transmembrane pressure of 0.1 bar using the set-up illustrated in Figure 1. The pressure of the system was monitored using a manometer that was connected to a vacuum air pump. The aeration rate was set in a range 0–1.5 L/min to evaluate its effect on permeability. The membrane panel was adjusted at a tilting angle of 0–20° to explore its effect on the fouling management.

### 2.4. Filtration Tests

The filtration tests were conducted to investigate the effect of two parameters: elevation/tilting angle and aeration rate. The elevation angles were set at 0° (vertical), 5°, 10°, 15°, and 20°. Higher tilting angles could not be done because of the system limitation. However, it was later found that the system reached plateau permeability at 15°. Plateau permeability is a maximum permeability obtained when reversible fouling is almost negligible, indicating maximum achievable value. For the test on the effect of tilting angle, the aeration rate was kept constant at 1.5 L/min. For the tests on the effect of aeration rate, the rates were set at 0, 0.25, 0.3, 1.0, and 1.5 (L/min) at a constant tilted angle of 15°. Each set of filtrations was run using a similar membrane panel to avoid membrane-to-membrane variation, which often occurs even for a sheet cut from the same batch of fabrication. The membrane was cleaned physically using detergent solution by gently wiping the surface using a soft sponge for about 5 min. The cleaning was extended if the clean water permeability could not be restored to >90% of the pristine value to ensure the membrane was at a comparable condition for each test. The filtration was done in a 10 min cycle consisting of 9 min filtration followed by 1 min relaxation. The permeate volume was collected during the relaxation period. For each set of parameters, the filtration was done in duplicate for 10 cycles corresponding to 100 min in which the queasy steady sate values of permeability were observed, indicated by <10% changes of permeability within the last three cycles of the filtration.

## 3. Results and Discussion

### 3.1. Effect of Tilting Angle

Figure 2 shows the filtration performance as a function of time, demonstrating that over the 10 cycles of the filtration, the permeability reached a queasy steady-state value, henceforth used as the parameter to evaluate the hydraulic performance of the filtration. Substantial decline in permeability in the earlier stage of filtration can be attributed to the irreversible adsorption fouling and pore blocking. At a higher tilting angle, gradual improvement in permeability with respect to higher panel tilting angles was observed (Figure 2). The permeability reached the maximum value of 221.4 ± 10.8 L/m^2^·h·bar at a 15° tilting angle, which corresponded to the plateau permeability. Plateau permeability means a point above which permeability does not increase by further adjustment of the panel tilting angle. Permeabilities of 221.4 ± 10.8, 212.0 ± 2.7, 204.2 ± 2.7, and 185.5 ± 2.7 L/m^2^·h·bar were obtained for panel tilting angles of 20°, 10°, 5°, and 0° respectively. This finding shows the impact of operating at an optimized tilting angle for better fouling management and improved permeability. Increase in permeability with tilting angle is attributed to the increase of contacts between the membrane surface and the air bubbles interaction. Higher contacts promote local mixing from the two-phase flow of liquid and air that disrupts foulant accumulation from the membrane surface. The possibility in achieving plateau permeance at a tilting angle of 15° also suggests that the maximum throughput can be achieved with a lower aeration rate at a higher angle to offer lower energy input.

The tilting panel system showed significant improvement in membrane fouling mitigation as well as hydraulic performance. Similar results were also reported by Eliseus et al. [23], who reported up to 1.7 times improvement in permeability for microalgae broth feed treatment by operating at a tilting angle of 20°. However, less improvement in permeability was experienced in the present research (about 19.4% higher than the vertical), which is attributed to the difference in the properties of the feed, the membrane, and, most importantly, the hydrodynamic conditions (aeration rate). It seems that the fouling propensity of the feed was not too severe, limiting the permeability gain obtained through aeration and panel tilting. 

### 3.2. Effect of Aeration Rate

Figure 3 demonstrates that the membrane permeability increased with increase in the aeration rate until reaching a plateau value of 199.6 ± 21.6 L/m^2^·h·bar. Thus, further increase in the aeration rate did not have an impact on the hydraulic performance. By operating at a 15° tilting angle and a 1 L/min aeration rate, an 83% improvement in permeability was obtained as compared to the same set-up under no aeration with a permeability of 109.1 ± 2.7 L/m^2^·h·bar. Thus, the finding justifies the positive effect of air bubbles in enhancing hydraulic performance for the membrane by imposing control of membrane fouling. It is worth noting that the permeability values obtained at an aeration rate of 1 L/min in Figure 3 varied largely in comparison to the rest of the data. It can be attributed to the randomness of measurement and was not addressed in great detail, particularly since it did not affect the trend on the effect of aeration rate on hydraulic performance. It is worth noting that substantial improvement in performance can be expected when optimum parameters and membranes materials are applied in the tilted panel system as demonstrated elsewhere [34,35].

### 3.3. Rejection and Detergent Recovery

The whiteness index of the cloth washed with fresh feed using 20 g detergent as per ASTM E313-05 was found to be 3.12 g, while the one washed with the reuse water was found to be 2.98 using 13.5 g detergent. The finding on the whiteness index of the cloth corresponds to 32% of detergent recovery. The analytical method of quantifying the detergent recovery was adopted from an earlier report by Giagnorio et al. [3]. They pioneered the quantitative method of estimating detergent recovery using ASTM E313-05. In their study, they found 40% detergent recovery, the maximum recovery reported thus far. These results justify the potential for reuse of treated laundry wastewater.

The membrane performance in terms of rejection of COD, TP, TN, and turbidity were evaluated, and the results are presented in Figure 4. The removal of turbidity, COD, TP, and TN was 77 ± 2.8%, 57 ± 0.8%, 30 ± 1.3%, and 6 ± 0.3%, respectively. Low TN rejection was obtained, since nitrogen containing molecules are relatively smaller than the membrane pore size and thus can pass through. The rejection of COD, TP, turbidity, and TN mostly originated from dirt particles bound by the detergent molecule. The free-detergent together with water permeate through the membrane pores, as reported by Šostar-Turk et al. [7]. The permeate can then be used as the make-up of fresh water and detergent. Similar results were reported by Manouchehri and Kargari [4]. They reported up to 75.5% and 98.4% rejection for COD and turbidity, respectively. The higher performance reported in comparison with the present research is attributed to the difference in the quality of the feed used, the pre-treatment, and the hydrodynamic conditions. However, by economic consideration of the operating conditions, the present research is more promising due to the system being operated at a lower pressure of 0.1 against 0.5 bar that was used by the other [4]. Sumisha et al. [9] also report high rejections when employing three polyether sulfone based ultrafiltration (UF) membranes for the treatment of laundry wastewater. The COD and the turbidity rejection of each membrane were evaluated; the one with the best performance was found to have COD and turbidity rejection of 88% and 98%, respectively. The higher performance observed is due to the superior performance of the modified UF membrane used in the research as compared to the MF used in the present research and also the higher turbulence induced by the stirred UF cell set-up [9]. 

Table 2 presents the comparison of the performance with some recently reported literature, showing that the present system offers less COD and turbidity removal. The process objective of the current work—detergent recovery—is different than the one in the references. Since the objective is to recover water and excess detergent for washing purposes, the excess detergent is detected in the permeate as COD as well as turbidity. On the other hand, the literature data aim either for water reuse (not for washing purposes or for discharge), which requires meeting certain quality standards.

## 4. Conclusions

This study demonstrates the potential of a tilted panel system for maximizing the impact of air bubble contact with the membrane surface to impose control in membrane fouling. The system provides a facile approach for water reuse and detergent recovery. The hydraulic performance increases by adjusting both aeration rate and tilting angle to reach the plateau permeability, suggesting most of the reversible fouling could be alleviated. At the plateau aeration value, permeability of 199.6 ± 21.6 (L/m^2^·h·bar) was obtained, which is 83% higher than the unaerated set-up. By tilting the membrane panel at 15° to the air bubbles, the permeability improved further to 221.4 ± 10.8 (L/m^2^·h·bar). The system also offers 32% detergent recovery. Overall findings suggest that the system offers an attractive approach for both fouling management and detergent recovery. 

## Figures and Tables

**Figure 1 membranes-10-00260-f001:**
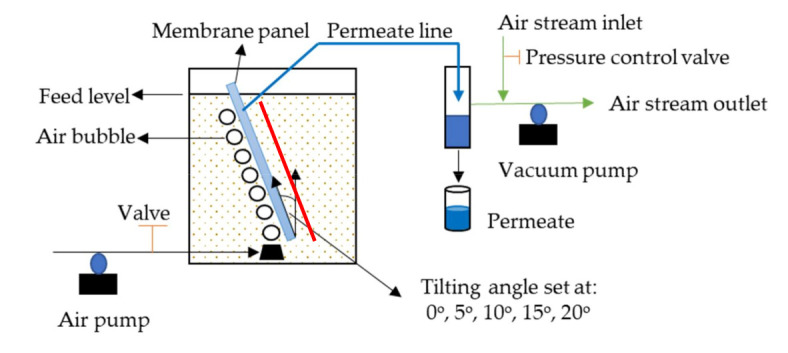
Experimental setup for laundry wastewater filtration using tilted panel system. The red line on the left side of the filtration panel indicates the membrane sheet fixed on the surface.

**Figure 2 membranes-10-00260-f002:**
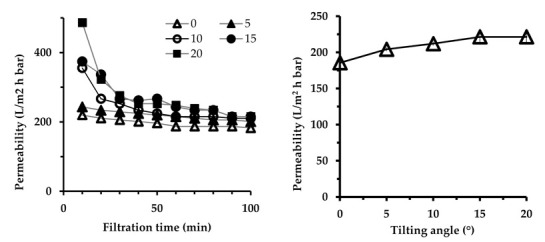
Effect of tilting angle on the steady-state permeability at constant aeration rate (1.5 L/min) as function of time (**left**) and the summary of the steady-state permeability values (**right**).

**Figure 3 membranes-10-00260-f003:**
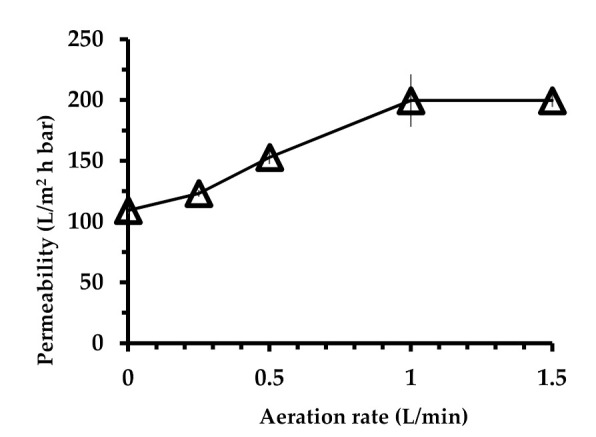
Effect of aeration rate on steady-state permeability at a constant 15° tilting angle.

**Figure 4 membranes-10-00260-f004:**
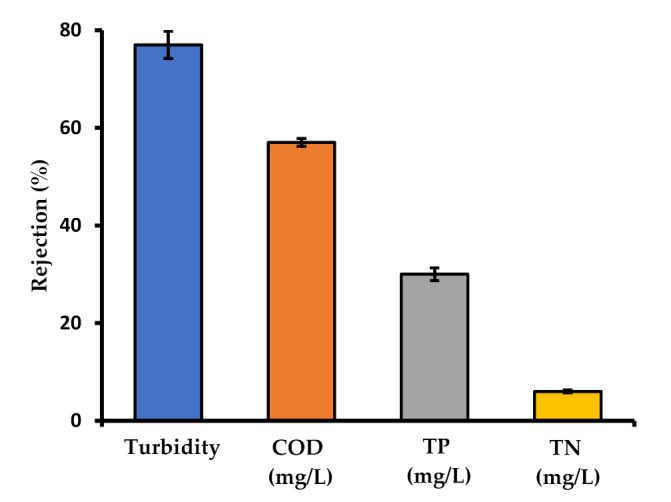
Rejection performance.

**Table 1 membranes-10-00260-t001:** Characteristics of the laundry wastewater sample.

Parameter	Laundry Wastewater Sample
Chemical oxygen demand (mg/L)	93.5
Total phosphorous (mg/L)	0.745
Total nitrogen (mg/L)	17.8
Turbidity (NTU)	2.185

**Table 2 membranes-10-00260-t002:** Performance of membrane filtration in laundry wastewater treatment.

Membranes	Pre-Treatment	COD Removal (%)	Turbidity Removal (%)	References
MF	None	57 ± 0.8	77 ± 2.8	This study
MF	COAG	61.0	100	[20]
UF	GACF	87.0	99.0	[27]
NF	None	97.0	98.0	[8]
MF	DF	90.0	98.4	[4]
UF	None	88.0	98.4	[9]

NF: nanofiltration, MF: microfiltration, UF: ultrafiltration, CAS: conventional activated sludge, COD: chemical oxygen demand, GACF: granular activated carbon filter, DF: 5 µm of polypropylene depth filter, MUF: modified ultrafiltration, COAG: coagulation.

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
