# Peer review of "Detergent and Water Recovery from Laundry Wastewater Using Tilted Panel Membrane Filtration System"

_membranes, 2020, doi:10.3390/membranes10100260_

Round 1
Reviewer 1 Report
The authors explored the use of air bubbles in a submerged filtration set-up as the means of inducing feed turbulence flow near the membrane surface for fouling mitigation, did good studies in effects of tilting angle and aeration rate on the steady-state permeability, and thus put forward an attractive approach for both fouling management and detergent recovery. This work is interesting, and effective membrane fouling control techniques can increase the system stability, and reduce the maintenance and operational cost. However, the authors should pay more attention to their grammar and presentation. Therefore, I recommend a revision to this work before its publication.
- Figures 3 and 4 should be rapainted, the format of axes in figure 3 should be changed and the quality of figure 4 should be improved.
- According to Table 2, this study had few advantages of COD removal and turbidity removal compared to other studies. It is suggested to make a better way for showing research values.
- Some sentences should be improved.
- More references could be refered, such as "doi.org/10.1016/j.cej.2020.125338" and "doi.org/10.1016/j.desal.2014.09.030"
Reviewer 2 Report
The article ID membranes-895399 is interesting, has scientific relevance presenting the effectiveness of submerged aerated tilted membrane panel for fouling management in real laundry wastewater filtration aimed for water and detergent reuse. However, this article presented here has gaps that need to be addressed so that it can be considered for publication in Membranes:
- Why was only one membrane made for use in this work?
-In the corrected version of the manuscript, add a table with the experimental conditions investigated in this work to facilitate the reader's understanding.
- The presentation of the schematic diagram in Figure 1 is confusing. Does the permeate current come out of the top or bottom of the membrane? Do the feed and permeate currents have any contact?
- Inline 168 there is a "." in duplicity.
- Why was the standard deviation of hydraulic permeability in section 3.2 higher than the other investigations? Add this information to the revised version of the article.
- Inline 207 there is a problem with the capital letter in the word They.
- Improve Figure 4 quality/resolution in the corrected version of the Review article.
- Was the membrane used in each experimental assay used? Or was a new membrane used in each experimental test? Add this information to the text.
- If the membrane was reused, do the cleaning procedures guarantee the recovery of the hydraulic permeability of the membrane for a new use?
Reviewer 3 Report
This manuscript by Barambu et al. investigates detergent and water recovery from laundry wastewater using a titled-panel membrane filtration system. Minimal characterization techniques were employed to support experimental observations. More needs to be done. A lot of information seems to be missing in the manuscript. While this work is rather interesting and has potential for application in the laundry industry, the explanations provided in most parts of the manuscript did not appear robust and were at times over-simplistic. I recommend that the following revisions be addressed before I reconsider the publication of this manuscript:
- It is in my humble opinion that while its content is generally coherent, this manuscript is not English-ready for publication. It needs to be thoroughly revised by a native English speaker for proper English language, grammar, punctuation, and overall style.
- In the Introduction section, please list the major pollutants (i.e., surfactant, oil/grease, suspended solids) that predominantly contribute to membrane fouling in laundry wastewaters. The entire section (especially the third paragraph) needs to focus on the major pollutants present in laundry wastewater that can be removed via membrane filtration systems. Therefore, the authors need to include more relevant texts pertaining to membrane fouling caused by surfactants and oils, and elaborate on fouling mitigation strategies targeting these foulants. The storyline needs to flow better in the Introduction section. From the second paragraph onwards, the focus should be on laundry wastewaters as well as surfactant- and oil-containing waste streams (since these are the potential foulants), not wastewaters in general.
- In lines 48–49, please list down the standards for the discharge and reuse of laundry wastewaters.
- In lines 52–53, membrane fouling also has a detrimental impact on membrane selectivity. The main focus of this manuscript seemed to be on flux enhancement while neglecting membrane selectivity. The authors ought to rectify this.
- In the third paragraph (line 56) of the Introduction section, the authors should write a few sentences focusing on how the major pollutants in laundry wastewaters cause fouling (e.g., electrostatic interaction, hydrophobic interaction) in membrane operations. Some relevant citations on fouling of membranes by surfactants and oils include:
- Polymeric ultrafiltration membranes and surfactants, Separation and Purification Reviews, 2003, 32(2), 215–278
- Surfactant effects on water recovery from produced water via direct-contact membrane distillation, Journal of Membrane Science, 2017, 528, 126–134
- Effect of surfactant hydrophobicity and charge type on membrane distillation performance, Journal of Membrane Science, 2019, 587, 117168
- Recent advances in membrane development for treating surfactant- and oil-containing feed streams via membrane distillation, Advances in Colloid and Interface Science, 2019, 273, 102022
- In lines 50–52, please add relevant citations focusing on the adoption of different membrane technologies for surfactant- and oil-containing wastewaters. Please also mention in the text the types of membrane processes that could potentially be applied:
Microfiltration/ultrafiltration
- The influence of surfactant on water flux through microfiltration membranes, Journal of Membrane Science, 1994, 86(3), 291–304
- Influence of adsorption and concentration of polarisation on membrane performance during ultrafiltration of a non-ionic surfactant, Desalination, 2002, 151, 21–31
Nanofiltration
- Surfactant fouling of nanofiltration membranes: Measurements and mechanisms, ChemPhysChem, 2007, 8, 1836–1845
- Evaluation of fouling mechanisms in the nanofiltration of solutions with high anionic and nonionic surfactant contents using a resistance-in-series model, Journal of Membrane Science, 2011, 367, 45–54
Reverse Osmosis
- Fouling behaviour of a reverse osmosis membrane by three types of surfactants, Journal of Water Reuse and Desalination, 2012, 2(1), 40–46
- Removal of fluorinated surfactants by reverse osmosis – Role of surfactants in membrane fouling, Journal of Membrane Science, 2014, 458, 111–119
Membrane distillation
- Wetting resistance of commercial membrane distillation membranes in waste streams containing surfactants and oil, Applied Sciences, 2017, 7(2), 118
- Polyvinylidene fluoride membrane modification via oxidant-induced dopamine polymerization for sustainable direct-contact membrane distillation, Journal of Membrane Science, 2018, 563, 31–42
- In line 61, please list and cite strategies for fouling management targeting surfactants and oils. Relevant citations on some other strategies include:
Membrane fabrication
- Application of sodium alginate-carrageenan coatings to PTFE membranes for protection against wet-out by surface-active agents, Separation Science and Technology, 2005, 40(5), 1067–1081
- Alginic acid-silica hydrogel coatings for the protection of osmotic distillation membranes against wet-out by surface active agents, Journal of Membrane Science, 2005, 260, 19–25
Membrane modification
- Superoleophobic surface modification for robust membrane distillation performance, Journal of Membrane Science, 2017, 541, 162–173
- Hierarchically structured Janus membrane surfaces for enhanced membrane distillation performance, ACS Applied Materials and Interfaces, 2019, 11, 28, 25524–25534
Air-bubbling
- Wetting preventing in membrane distillation through superhydrophobicity and recharging an air layer on the membrane surface, Journal of Membrane Science, 2017, 530, 42–52
- In the Results and Discussion section, please include SEM images (surface and cross-section) and describe the morphology of the membranes. Please also include data on the pore size distribution of the membranes. The authors need to dedicate a section to discuss on membrane characterization results. More membrane characterization techniques (e.g., zeta potential, roughness) need to be employed. In addition, the authors need to provide more information on the properties of the laundry wastewater such as pH, zeta potential, conductivity, surface tension, and TSS. What is/are the surfactant(s) present? Anionic/cationic/non-ionic? Surfactant concentration? CMC value? HLB value? All these factors could potentially affect membrane fouling.
- In line 169, I believe the authors meant 15°, 10°, 5°, and 0°. How long were the filtration experiments conducted for? How many replicates were performed for each condition? Did the authors perform membrane autopsy on the membranes after these experiments? Improved permeability does not necessarily mean better fouling management. The readers need to see concrete evidence of fouling. What about the selectivity/rejection of the membranes for each experiment? The authors need to report the rejection values for each condition. More importantly, the authors need to perform a control experiment without tilting AND aeration for a fair comparison. The authors should include the permeability vs time and TMP vs time profiles for every experiment.
- Where are the UV-vis data for surfactant concentration?
- Is there a reason why the authors decided to use those membranes in this study?
Round 2
Reviewer 3 Report
I'm satisfied with the response provided by the authors.